# Minimum Winter Temperature as a Limiting Factor of the Potential Spread of *Agrilus planipennis*, an Alien Pest of Ash Trees, in Europe

**DOI:** 10.3390/insects11040258

**Published:** 2020-04-23

**Authors:** Marina J. Orlova-Bienkowskaja, Andrzej O. Bieńkowski

**Affiliations:** A.N. Severtsov Institute of Ecology and Evolution, Russian Academy of Sciences, 119071 Moscow, Russia; bienkowski@yandex.ru

**Keywords:** emerald ash borer, EAB, Norway, Finland, Sweden, European Russia, low temperature

## Abstract

The emerald ash borer, EAB (*Agrilus planipennis*) is a devastating alien pest of ash trees. It is spreading in European Russia and Ukraine and will appear in other European countries. Our aim was to determine the regions of Europe where the winter temperature drops low enough to prevent *A. planipennis* establishment. We calculated the minimum daily air temperature from 2003–2019 for each grid square (0.5° × 0.5°) in East Asia, North America and Europe and determined the minimum daily temperature in the grid squares where *A. planipennis* was recorded. Temperatures of −30 to −33 °C occur in the northern portions of the pest range on all continents. No established population has been recorded in localities where temperatures below −34 °C occur. This temperature is close to the absolute supercooling point of *A. planipennis* larva (−35.3 °C). It is unlikely that low temperatures could prevent the spread of *A. planipennis* in northern Western Europe (Sweden, Norway, Finland, etc.), since the temperature in this area did not fall to −34 °C from 2003–2019. However, such temperatures are not rare in eastern European Russia (Kostroma, Vologda, Orenburg regions, etc.), where *Fraxinus pennsylvanica* and *F. excelsior* occur. These regions could potentially become refuges for these ash species.

## 1. Introduction

The emerald ash borer (*Agrilus planipennis* Fairmaire) (Coleoptera: Buprestidae) is a devastating alien pest of ash trees in European Russia and North America [1,2,3]. The native range of this wood-boring beetle occupies a restricted territory in East Asia, namely in northeastern China, Japan, the Korean Peninsula and the southern Russian Far East [4]. Since the first record in America in 2002, the pest has spread to 35 states of the USA and five provinces of Canada and killed countless millions of ash trees [5]. In 2003, *A. planipennis* was first recorded in Europe in Moscow, and a severe outbreak and quick spread of the pest began [1,3]. By 2020, *A. planipennis* had spread to 16 regions of European Russia and to the east of Ukraine [6,7].

*Agrilus planipennis* is continuing to spread and will soon appear in other European countries [8,9]. It presents a serious threat to ash trees all over Europe [3,10], so it is included on the list of 20 priority quarantine pests of the EU [11]. It is very important to determine the potential range of *A. planipennis* in Europe and the factors that could limit the spread of the pest. This information is crucial for quarantine protocols and for the elaboration of measures of pest control. In addition, it is necessary to reveal the regions where ash trees occur but where the climate is not suitable for *A. planipennis*. It is estimated that *A. planipennis* will eventually decimate nearly all ash (*Fraxinus* spp.) in North America [12]. *Agrilus planipennis* is also dangerous for ash trees all over Europe [10]. The regions where the winter is too cold for *A. planipennis* could potentially become refuges for ash species.

The spread of *A. planipennis* in Europe could be limited by the native distribution of its host plants: *Fraxinus* spp. [10] However, it should be taken into account that ash trees are often planted in cities far beyond the native ranges of ash species. In particular, *F. pennsylvanica*, which was introduced from North America and is highly susceptible to *A. planipennis* [13], is one of the most popular trees planted in the cities of European Russia and some other European countries [14]. The high-resolution map of *Fraxinus* spp. distribution in Central Europe is available [15]. However, no such map or city tree inventories for European Russia are available. An additional biotic factor that could theoretically limit the spread of *A. plenipennis* is the spread of its natural enemies, e.g., the parasitoid *Spathius polonicus* Niezabitowski [16], but the influence of this factor has not yet been studied.

The only attempt to predict the potential *A. planipennis* range in Europe by climatic variables was conducted using maximum entropy modelling [17]. This model has a number of serious flaws. First, the survey data from Orlova–Bienkowskaja [18] used as a base for this model were misunderstood: all surveyed locations were regarded as locations of *A. planipennis* detection, although in fact, the surveys in many locations indicated negative results. Second, the true range of *A. planipennis* by 2020 was more extensive than the output of this model [7].

It is yet unknown what climatic factors could limit the spread of *A. planipennis* to the north in Europe. For alien insects, low winter temperatures are one of the most important abiotic factors limiting northern range edges [19]. A study conducted in North America has shown that cold winter temperatures could potentially limit the spread of *A. planipennis* in the USA and Canada [20,21]. Therefore, it is important to determine the regions of Europe where winter cold could limit the spread of *A. planipennis*.

The life cycle of *A. planipennis* is 1 year in warm regions (specifically in Tianjin, China) [22], 2 years in cold regions (specifically in Moscow, Russia) [23] and 1 or 2 years in regions with an intermediate climate (specifically, in New York, NY, USA) [24]. With a 1-year life cycle, *A. planipennis* overwinters as a J-larva. With a 2-year life cycle, *A. planipennis* overwinters twice: the first time as a larva in a larval gallery under the bark and the second time as a J-larva in a pupal cell in the sapwood.

The J-larvae survive low temperatures by accumulating high concentrations of glycerol in their body fluids and by synthesizing antifreeze agents [25]. The supercooling point of the overwintering J-larvae of *A. planipennis* is different in different regions. The absolute minimum value that has been recorded experimentally was −35.3 °C [25]. J-larvae have not been recorded to survive temperatures below this value. These experimental data are supported by observations in nature in North America. In particular, severe mortality (93%) of the overwintering larvae was caused by low temperatures (below −30 °C and up to almost −40 °C) on the 13–15 February 2016 in Syracuse, New York, NY, USA [26].

We determined the minimum mean daily temperatures in localities with *A. planipennis* in the last 17 years (i.e., after the first record of *A. planipennis* in European Russia). This allowed us to determine the minimum temperature that did not prevent the survival of *A. planipennis* populations. Then, we assessed the regions of Europe where winter temperature could potentially prevent the spread of the pest.

## 2. Materials and Methods

We compiled a table of occurrences of *A. planipennis* in Asia, North America and Europe (Appendix A) using information from current databases and articles [4,5,7,27].

The beneath-bark temperature changes slower than the air temperature [20]. Therefore, we used the mean daily temperature to level the short-term air temperature fluctuations. The initial data on air temperature for each day from 1 January 2003 through 1 October 2019 were obtained from the ERA5-Land Global Atmospheric Reanalysis dataset for each 0.5° × 0.5° grid in Europe (from 30 to 75° N and from 20° W to 70° E), Asia (from 20 to 60° N and from 100 to 150° E) and North America (from 10 to 65° N and from 50 to 140° W) [28]. These squares covered the whole current range of the pest in the world and all of Europe. We did not take into account the record from the Xinjiang Uygur Autonomous Region, China. It was unclear if *A. planipennis* had established in this region, since the number of found specimens and (or) damaged trees was unknown [4].

The daily mean temperature was calculated for each day in each grid square as an average of the temperatures at 00:00, 02:00, 04:00, 06:00, 08:00, 10:00, 12:00, 14:00, 16:00, 18:00, 20:00 and 22:00 UTC. A detailed description of the calculation method and the computer code used is provided in the Appendix A. Then, the minimum mean daily temperature recorded from 2003–2019 was determined for each grid (Appendix A). Visualization of the minimum temperature and localities of *A. planipennis* occurrence in the same map allowed us to compare these data. The visualization was created using DIVA-GIS 7.5 [29].

Then, we identified the days where the daily average temperature was below −30 °C in each grid square where *A. planipennis* was detected (Appendix A). This allowed us to determine the daily air temperatures in which the pest populations can survive.

The temperature experienced by the overwintering J-larva could slightly differ from the air temperature in the region. First, the grids in the ERA5 Global Atmospheric Reanalysis dataset were 0.5° × 0.5° [28]. Therefore, the temperature in a particular locality could slightly differ from the temperature calculated for the grid. In particular, in the center of large cities, the temperature could be 1–2 °C higher than that in the surrounding territories [30]. Second, when the nightly low temperatures occurred, temperatures beneath the bark were typically about 1 °C warmer than nearby air temperatures [20]. Besides that, the location of a tree (e.g., proximity to buildings, southern exposure and density of neighboring trees) could affect low temperatures that a tree might experience [20]. We did not measure the temperature experienced by individual larvae but used the mean temperature of the coldest day as an index of the severity of winter cold in the region. Our model did not take into account global climate change.

## 3. Results

### 3.1. Temperatures below −30 °C in the Native Range of A. planipennis in Asia

Winter temperatures below −30 °C are not rare in the northern part of the native range in the Russian Far East and in northeastern China (Figure 1). From 2003–2019, days with mean daily temperatures below −30 °C were recorded in at least 14 localities of Asia where *A. planipennis* was recorded during this period (Table 1). The coldest days with mean temperatures below −33 °C were recorded in Troitskoe and Khabarovsk city (Russia) and in Yichun (China).

No direct data on the percentage of larvae that survived the low temperature in Asian *A. planipennis* populations were available. However, a comparison of the pest distribution with the climatic data showed that the populations lived in localities where a very low temperature was usual. For example, a population of *A. planipennis* in Khabarovsk was first recorded in 2006 and still existed in 2016 despite temperatures below −30 °C occurring there every year from 2010 to 2014, and the minimum mean daily temperature being below −33 °C [31].

The mean daily temperature did not fall to −34 °C in any *A. planipennis* locality in Asia from 2003–2019.

### 3.2. Temperatures below −30 °C in the Current Range of A. planipennis in North America

A similar analysis of climatic data made for the current range of *A. planipennis* in North America (Figure 2) has shown that very cold days with mean temperatures below −30 °C were recorded in several *A. planipennis* localities from 2003–2019 (Table 2). Temperatures of up to at least −31 °C did not destroy *A. planipennis* populations in North America. For example, a population in St. Paul (Minnesota, USA) (locality 5 in Figure 2) was recorded in 2009 [5] and still exists despite the low temperatures in January 2019 (−31.10 °C, see Appendix A) [27].

There are still no recorded cases of survival of *A. planipennis* on days with average temperatures below −34 °C in North America. The only locality of detection of *A. planipennis* in which the minimum mean daily temperature recorded from 2003–2019 was below −34 °C, was Winnipeg (Canada). However, *A. planipennis* was first detected only three years ago: in November 2017 [42]. In January 2019 a temperature of −35.36 °C was recorded there (see Appendix A). Then, in summer 2019 no adults of *A. planipennis* were detected in Winnipeg [43] Therefore, it is unknown whether the population is stable or transient.

### 3.3. Temperatures below −30 °C in the Current Range of A. planipennis in European Russia

The days with mean temperatures below −30 °C occurred in most localities of *A. planipennis* in European Russia from 2003–2019 (Figure 3, Table 3). However, no cases of extinction or decline of populations after severe cold have been recorded. In particular, in January 2006, temperatures below −30 °C were recorded in the Moscow region. However, despite this, a severe outbreak and quick spread of *A. planipennis* was recorded in the Moscow region from 2006–2009 [1,44,45,46].

The northernmost European population of *A. planipennis* was discovered in 2013 in the city of Yaroslavl [18]. This population survived despite frosts in January 2017: the mean daily temperature was −33.76 °C on 7 January and −33.69 °C on 8 January. Moreover, the range of *A. planipennis* in Yaroslavl expanded: in summer 2019, *A. planipennis* was recorded in 11 new localities and killed hundreds of trees in the city [47].

No populations of *A. planipennis* have been recorded in European Russia in regions where mean daily temperatures below −34 °C occurred from 2003–2019.

## 4. Discussion

The data from Asia, North America and European Russia indicated that populations of *A. planipennis* could survive in regions where mean daily temperatures below −30 °C occurred. Even mean daily temperatures of −33 °C did not destroy *A. planipennis* populations. The winters in northern Western Europe were warmer than those in the current northern range of *A. planipennis* in European Russia and in the northern native range in Asia (Figure 4). Therefore, it is unlikely that winter cold could prevent the spread of *A. planipennis* in Norway, Sweden, Finland and other European countries.

Established populations of *A. planipennis* have not been recorded in regions with severe winters where days with mean temperatures below −34 °C occur. According to the experimental data, such temperatures cause high mortality in the overwintering J-larvae [25,48]. The absolute supercooling point, i.e., the minimum temperature at which *A. planipennis* J-larvae could survive, is −35.3 °C [21].

Temperatures below −34 °C are not rare in eastern European Russia (see Figure 4 and Appendix A). Therefore, winter cold could probably become the limiting factor of *A. planipennis* spread to this region. In particular, it is unlikely that *A. planipennis* will establish in the Arkhangelsk region, Republic of Komi, most of the Vologda region, Kostroma region, Kirov region, Udmurt Republic, Republic of Bashkortostan or most of the Orenburg region. Temporary populations of *A. planipennis* could theoretically appear in these regions in the case of several subsequent warm winters, but these populations would probably be destroyed by the next cold winter.

*Fraxinus pennsylvanica* is very common in urban and roadside plantings in eastern European Russia. It has become an established nonnative plant in the Orenburg region and Republic of Bashkortostan and occurs in natural habitats there [49]. This ash species is very cold-tolerant. In particular, plantings of *F. pennsylvanica* in Ufa (Bashkortostan) survived extremely cold winters when the temperatures fell below −40 °C [50].

*Fraxinus excelsior* is also sometimes planted in the cities of eastern European Russia. In addition, it is native in the Kostroma, Vologda and Orenburg regions and occurs in the forests there [51,52,53]. These regions, where the winters are too cold for *A. planipennis* but the climate is suitable for *F. excelsior* and *F. pennsylvanica*, could potentially become a refuge for these ash species.

It is interesting that the range of *A. planipennis* in European Russia has not expanded to the north since 2013, although the range has significantly expanded to the south [7]. This additional indirect evidence suggests that the current northernmost *A. planipennis* locality (Yaroslavl) is close to the northern border of the potential range.

We used the data since 2003, because *A. planipennis* was first recorded in Europe in 2003. Little is known about the distribution of *A. planipennis* in Asia before records in North America and European Russia. There is no evidence that *A. planipennis* survived the temperatures below −34°C in its native range before 2003.

The minimum winter temperature is important but probably not the only factor determining the potential *A. planipennis* range. The possible impact of accumulated degree days during the growing season and other factors should be studied in the future. It would also be interesting to make direct year-round measurements of the temperature under the bark of ash trees in Moscow, Yaroslavl and other localities of *A. planipennis* in European Russia. This would allow us to determine what temperatures are experienced by the larvae and J-larvae in nature.

## 5. Conclusions

Populations of *A. planipennis* survive in regions where the minimum mean daily temperature is −30 to −33 °C.No established populations of *A. planipennis* have been recorded in regions where the minimum mean daily temperature falls below −34 °C.It is unlikely that winter temperature would become a limiting factor preventing the spread of *A. planipennis* in Norway, Sweden, Finland or other countries of Western Europe.Low winter temperature could become the limiting factor that prevents the spread of *A. planipennis* to eastern European Russia, where daily average temperatures below −34 °C are not rare.*Fraxinus pennsylvanica* and *F. excelsior* occur in eastern European Russia, where the winters are too cold for *A. planipennis*. Therefore, eastern European Russia could potentially become a refuge for these ash species.

## Figures and Tables

**Figure 1 insects-11-00258-f001:**
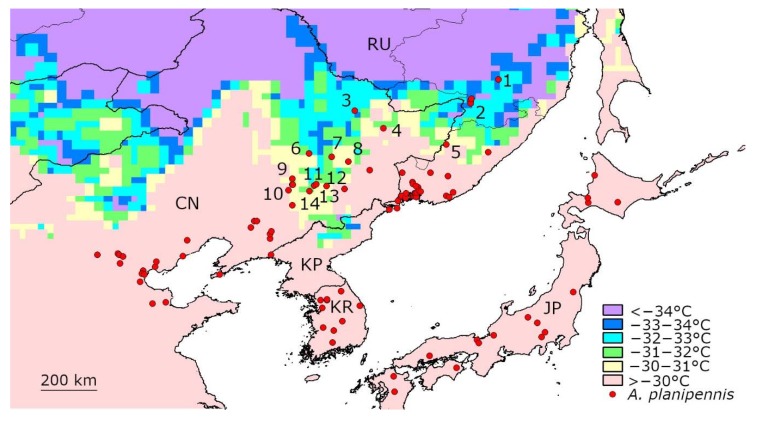
Mean temperature of the coldest day recorded from 2003 to 2019 in the native range of *A. planipennis* in Asia. Some localities where the daily temperature fell below −30 °C were numbered. Detailed information for each of these localities is provided in Table 1. CN—China, JP—Japan, KP—Democratic People’s Republic of Korea, KR—Republic of Korea, RU—Russia.

**Figure 2 insects-11-00258-f002:**
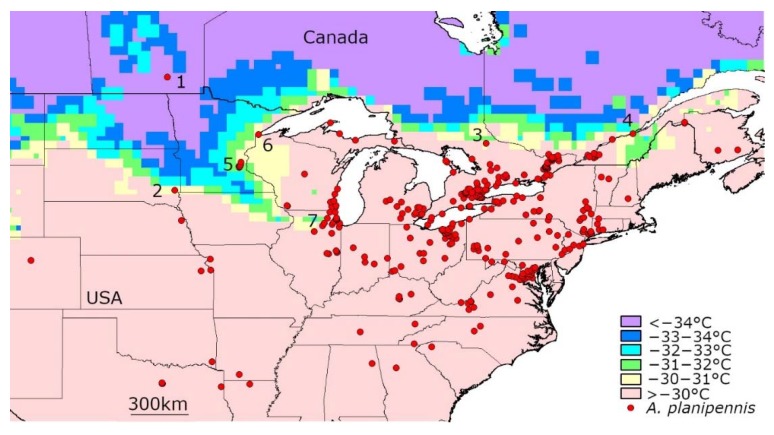
Mean temperature of the coldest day recorded from 2003 to 2019 in the territories currently occupied by *A. planipennis* in North America. Some localities where the daily temperature fell below −30 °C are numbered. The detailed information for each of these localities is provided in Table 2.

**Figure 3 insects-11-00258-f003:**
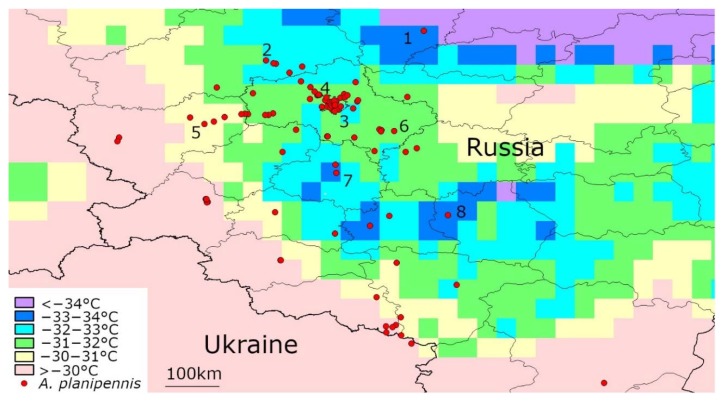
Minimum mean daily temperatures recorded from 2003 to 2019 in the territories currently occupied by *A. planipennis* in Europe (Russia and Ukraine). Some localities where the daily temperature fell below −30 °C are numbered. Detailed information for each of these localities is provided in Table 3.

**Figure 4 insects-11-00258-f004:**
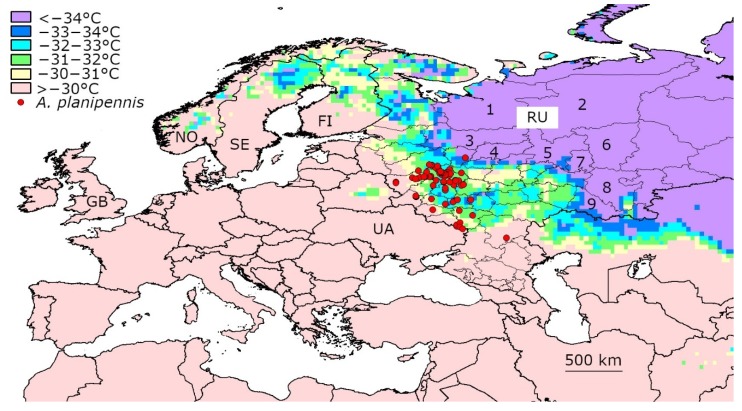
Minimum mean daily temperatures recorded from 2003 to 2019 in northern Europe. FI—Finland, GB—Great Britain, NO—Norway, RU—Russia, SE—Sweden, UA—Ukraine. Regions east of European Russia are numbered: 1—Arkhangelsk region, 2—Komi Republic, 3—Vologda region, 4—Kostroma region, 5—Kirov region, 6—Perm region, 7—Udmurt Republic, 8—Republic of Bashkortostan, 9—Orenburg region.

**Table 1 insects-11-00258-t001:** Temperatures below −30 °C occurred from 2003–2019 in the localities where *A. planipennis* populations were recorded during this period *.

Country	Locality (Numbers Correspond to the Numbers in Figure 1)	Year of *A. planipennis* Detection	Number of Days with the Mean t° below −30 °C	Years When the Days with the Mean t° below −30 °C Occurred	Mean Temperature of the Coldest Day (°C)
Russia	Troitskoe 1	2003–2008	28	2005, 2006, 2010, 2011, 2012, 2013, 2014, 2017	−33.86
China	Yichun 3	2003	17	2006, 2010, 2011, 2012, 2013, 2014, 2018	−33.19
Russia	Khabarovsk 2	2004–2006, 2016	15	2010, 2011, 2012, 2013, 2014	−33.12
China	Harbin 6	1964, 2003–2008	11	2010, 2011, 2012, 2013, 2018	−32.28
China	Laoshan 7	2003–2008	4	2011, 2018	−31.91
China	Jiang Nan Forest 11	2006	4	2009, 2010, 2011, 2018	−31.77
China	Songhuahu Nature Reserve 14	2003	2	2010, 2011	−31.69
China	Jilin City 13	2003–2008	2	2010, 2011	−31.69
China	Jiutai 9	2003–2008	2	2011, 2012	−31.41
China	Hejiang 4	2003	4	2011, 2013	−31.35
Russia	Dalnerechensk 5	2004–2006	4	2011, 2012, 2013	−31.23
China	Jiaohe Experimental Forest 12	2003	3	2011, 2018	−30.75
China	Shangzhi 8	2003, 2012	1	2018	−30.67
China	Changchun 10	2003, 2008	2	2011, 2012	−30.16

* Detailed information about the coordinates of the localities, temperatures and dates is provided in Appendix A. Sources of information about *A. planipennis* detection: [32,33,34,35,36,37,38,39,40,41].

**Table 2 insects-11-00258-t002:** Some *A. planipennis* localities in North America where the average temperature of the coldest day from 2003–2019 was below −30 °C *.

Country	Locality (Numbers Correspond to the Numbers in Figure 2)	Year of *A. planipennis* Detection	Number of Days with the Mean t° below −30 °C	Years When the Days with the Mean t° below −30 °C Occurred	Mean Temperature of the Coldest Day (°C)
Canada	Manitoba, Winnipeg 1	2017	35	2004, 2005, 2007, 2008, 2009, 2013, 2014, 2015, 2016, 2019	−37.75
Canada	Quebec, L’Islet 4	2018	1	2004	−31.15
USA	Minnesota, St. Paul 5	2008	2	2019	−31.10
USA	Illinois, Kane 7	2006	1	2019	−31.10
USA	South Dakota, Minnehaha 2	2018	1	2009	−30.36
USA	Wisconsin, Douglas 6	2013	1	2019	−30.15
Canada	Ontario, Renfrew 3	2013	1	2004	−30.03

* Detailed information about the coordinates of localities, temperatures and dates is provided in Appendix A. Sources of information about *A. planipennis* detection: [5,27].

**Table 3 insects-11-00258-t003:** Some *Agrilus planipennis* localities in European Russia where the mean temperature of the coldest day from 2003–2019 was below −30 °C *.

Region	Locality (Numbers Correspond to the Numbers in Figure 3)	Year of *A. planipennis* Detection	Number of Days with the Mean t° below −30 °C	Years When the Days with the Mean t° below −30 °C Occurred	Mean Temperature of the Coldest Day (°C)
Yaroslavl	Yaroslavl 1	2013–2019	7	2006, 2012, 2017	−33.76
Tambov	Michurinsk 8	2013, 2017	5	2006, 2010	−33.64
Tula	Tula 7	2013, 2019	3	2006	−33.14
Tver	Tver 2	2015–2019	4	2006, 2017	−32.03
Moscow	Moscow 3	2003–2019	2	2006	−31.94
Moscow	Zelenograd 4	2011–2019	3	2006, 2017	−31.86
Moscow	Kolomna 6	2012, 2017	1	2006	−31.54
Smolensk	Vyazma 5	2012, 2019	2	2006	−30.95

* Sources of information about records of *A. planipennis*: [1,7,8,18].

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
