# Peer review of "Minimum Winter Temperature as a Limiting Factor of the Potential Spread of *Agrilus planipennis*, an Alien Pest of Ash Trees, in Europe"

_insects, 2020, doi:10.3390/insects11040258_

Round 1
Reviewer 1 Report
The most important take-home of this analysis is not where it is too cold for EAB, because there is much uncertainty in the model, but rather where EAB is likely to be able to establish, i.e. most of Europe. European countries would do well to prepare for an eventual invasion.
Other than some mentions in the discussion about locations where ash occurs but EAB doesn’t, the authors do not consider the distribution of ash in their model. This is a glaring omission. If there are national forestry/urban tree databases that could be accessed and added to the model it would make the conclusions much more robust. Without that analysis, the model is much less useful.
Specific comments are as follows:
Title: The authors should not refer to low temperatures as a “frost”. A frost occurs when visible frost appears on surfaces and depends on the moisture content in the air as well as the temperature. A freeze occurs when air temperature drops below freezing. You can get a frost when temperatures are above freezing and a freeze without frost. What is limiting to EAB is minimum cold temperatures in winter. So perhaps the title should read “Winter temperatures as a limiting factor…………..”.
Line 13: The authors could say …where winter temperatures drop low enough to prevent A. planipennis establishment”. But do not use the word “frost” in this manuscript.
Line 42: The authors correctly consider that the distribution of the host tree (ash) could affect the distribution of EAB as one moves north. However, in the analysis, no attempt is made to consider whether ash occurs in a given grid. In the United States, at least, there are forest and city tree inventories that would show whether ash is present. For the model to be predictive based on temperature on needs to make sure that all areas being modeled have ash but those without EAB are too cold.
Line 66: Technically EAB in overwintering chambers are still fourth instar larvae referred to as J-larvae. If disturbed, these larvae can continue feeding and can re-form an overwintering chamber. They do not become true pre-pupae (which do not feed or move) until the spring when they lose the J-shape and become short and fat. Overwintering EAB should be referred to as J-larvae.
Line 84: Using the average daily temperature to account for the lag time for beneath-bark temperatures is an interesting idea. Do the authors have any evidence that temperatures under the bark do not get as cold as the air temperature or do they get just as cold only later? This may not be all that important, however, because it is a very fine detail about the temperature experienced by the insects when the model is based on a much more course grained estimate of temperatures. The model is based on the minimum average daily temperature over 17 years over a large area (grid). Temperatures over time and across space can vary considerably. It is conceivable that EAB could establish somewhere for a while only to be pushed back due to an extreme weather event. But then they are likely to rebound given that 100% mortality is unlikely to occur due to microclimate effects.
Line 183: The authors state that the mean temperature of the coldest day could serve as an indicator of the severity of winter temperatures, but they do not provide evidence.
Line 216: The authors keep reiterating that northern Russia could be a refuge for Fraxinus excelsior and Fraxinus pennsylvanica. Why is this important, especially given that F. pennsylvanica is not native?
Line 222: The authors do not discuss climate change, but they should. Climate change has resulted not only in overall warmer winters, which could lead to an expansion of EAB distribution, but also to more frequent extreme events such as bomb cyclones and polar vortexes due to a weakening of the jet stream. These two possible scenarios should be discussed.
Author Response
- Thank you very much for the thorough review of our article and the very good advices.
Title: The authors should not refer to low temperatures as a “frost”. A frost occurs when visible frost appears on surfaces and depends on the moisture content in the air as well as the temperature. A freeze occurs when air temperature drops below freezing. You can get a frost when temperatures are above freezing and a freeze without frost. What is limiting to EAB is minimum cold temperatures in winter. So perhaps the title should read “Winter temperatures as a limiting factor…………..”.
- Thank you. The title is corrected according to your advice. : Line 2: Minimum winter temperatures as a limiting factor of the potential spread of Agrilus planipennis, an alien pest of ash trees, in Europe
Line 13: The authors could say …where winter temperatures drop low enough to prevent A. planipennis establishment”. But do not use the word “frost” in this manuscript.
- It is corrected through the manuscript: the word “frost” is replaced by “low temperature”: Lines 13, 17, 18, 20, 22, 26, 80, 85, 121, 125, 126, 132, 134, 135, 149, 153, 156, 159, 164, 176, 179, 183, 201, 215, 256, 259, 268.
Line 42: The authors correctly consider that the distribution of the host tree (ash) could affect the distribution of EAB as one moves north. However, in the analysis, no attempt is made to consider whether ash occurs in a given grid. In the United States, at least, there are forest and city tree inventories that would show whether ash is present. For the model to be predictive based on temperature on needs to make sure that all areas being modeled have ash but those without EAB are too cold.
- It is really a glaring omission, but not ours. The information on ash distribution as well as on EAB distribution in Europe is much-much poorer than in the United States. No city trees inventories are available for European Russia. The high-resolution map of ash distribution is available only for Central Europe (Valenta et al., 2015). We plan to make the map of ash trees in Europe using satellite images of the surface of the earth. The explanation is added to the Lines 52, 53: Unfortunately no city tree inventories of European Russia are available, and the high-resolution map of ash distribution is available only for Central Europe.
- By the way, I envy greatly when I see Emerald ash borer info website. Thousands of people contribute to this North American inventory. Unfortunately, less than 10 people study A. planipennis distribution in Russia. And none of them is employer of National Plant Protection Organisation. Unfortunately, NPPO of Russia and Ukraine are trying to conceal the real EAB range.
Line 66: Technically EAB in overwintering chambers are still fourth instar larvae referred to as J-larvae. If disturbed, these larvae can continue feeding and can re-form an overwintering chamber. They do not become true pre-pupae (which do not feed or move) until the spring when they lose the J-shape and become short and fat. Overwintering EAB should be referred to as J-larvae.
- Thank you. “Pre-pupae” are changed to “J-larvae” all over the manuscript.
Line 84: Using the average daily temperature to account for the lag time for beneath-bark temperatures is an interesting idea. Do the authors have any evidence that temperatures under the bark do not get as cold as the air temperature or do they get just as cold only later?
- Venette and Abrahamson have studied the beneath-bark temperatures of ash trees [https://www.nrs.fs.fed.us/disturbance/invasive_species/eab/control_management/cold_hardiness/bp-EAB-and-extreme-cold.pdf]. They write: “The bark does offer some insulation to the larva from nightly low temperatures. When these lows occur, temperatures beneath the bark are typically about 1°C (about 2°F) warmer than nearby air temperatures.” Therefore, the short-term extreme colds are less important for the larvae than daily temperatures. We have corrected Materials and Methods to clarify this idea: Line 116.
This may not be all that important, however, because it is a very fine detail about the temperature experienced by the insects when the model is based on a much more course grained estimate of temperatures. The model is based on the minimum average daily temperature over 17 years over a large area (grid). Temperatures over time and across space can vary considerably. It is conceivable that EAB could establish somewhere for a while only to be pushed back due to an extreme weather event. But then they are likely to rebound given that 100% mortality is unlikely to occur due to microclimate effects.
- Theoretically, it is conceivable that planipennis population could survive and rebound after the air temperature –34 °C. But no such cases have been recorded. In fact, even the short-term fall of temperature to –30 °C cause very high EAB mortality (93%) (Jones et al, 2017). We do not state that larvae in the cold regions are killed by just the extreme temperature –34 °C. We just state that no established populations have been recorded in the regions, where the daily temperature in the region sometimes falls below –34 °C.
Line 183: The authors state that the mean temperature of the coldest day could serve as an indicator of the severity of winter temperatures, but they do not provide evidence.
- We added the explanation to Materials and Methods to make our idea clearer: Lines 109-122.
Line 216: The authors keep reiterating that northern Russia could be a refuge for Fraxinus excelsior and Fraxinus pennsylvanica. Why is this important, especially given that F. pennsylvanica is not native?
- “It is estimated that EAB will eventually decimate nearly all ash (Fraxinus spp.) in North America” (DeSantis et al., 2013). And it is quite possible that EAB will eventually decimate nearly all ash in Europe. Therefore, it is important to reveal in what regions of the world F. pennsylvanica, F. excelsior and other Fraxinus species could survive. Such potential refuges were revealed in the north of the United States (DeSantis et al., 2013). Our study has shown that in the north-east of European Russia, where Fraxinus spp. grows, A. planipennis cannot survive. The explanation is added: Lines 43-46.
Line 222: The authors do not discuss climate change, but they should. Climate change has resulted not only in overall warmer winters, which could lead to an expansion of EAB distribution, but also to more frequent extreme events such as bomb cyclones and polar vortexes due to a weakening of the jet stream. These two possible scenarios should be discussed.
- Thank you. We have discussed two scenarios according your recommendations: Lines 239-234.
The most important take-home of this analysis is not where it is too cold for EAB, because there is much uncertainty in the model, but rather where EAB is likely to be able to establish, i.e. most of Europe. European countries would do well to prepare for an eventual invasion.
- Thank you for this comment. Our model definitely shows that the climate in the east of European Russia is not suitable for planipennis. For example, in the city of Kostroma, where F. pennsylvanica is usual all over the city, the temperature below -35 °C is usual in winter. No cases of survival of EAB populations in such climate have been yet recorded.
Other than some mentions in the discussion about locations where ash occurs but EAB doesn’t, the authors do not consider the distribution of ash in their model. This is a glaring omission. If there are national forestry/urban tree databases that could be accessed and added to the model it would make the conclusions much more robust. Without that analysis, the model is much less useful.
- Thank you for this comment. It would be really useful to add the information about the distribution of Fraxinus in Europe including European Russia and provide a map. But the relevant map or database does not exist. There are very few data on Fraxinus spp. Russia in Global Biodiversity Information Facility and in European Forest Genetic Resources Programme. The information on some other European countries is also insufficient. In fact, the good map of ash distribution has been made only for Central Europe (Valenta, V., Moser, D., Kuttner, M., Peterseil, J., & Essl, F. (2015). A high-resolution map of emerald ash borer invasion risk for southern central Europe. Forests, 6(9), 3075-3086.). We plan to make a high-resolution map of Fraxinus spp. range in Europe in cooperation with the experts in remote earth sensing and thematic deciphering in ecology. Compilation of this map is a large and complicated work which cannot be included in the current article.
- We are really very grateful to you for the clever questions and wish you to stay health!
Reviewer 2 Report
More specific comments in detail as follows:
1. In the "Materials and Methods", it was described that "……Asia (from 20 to 60 °N and from 100 to 150 °E) and……These squares covered the whole current range of the pest in the world and all of Europe". Emerald ash borer was found to occur in Xinjiang-Uygur Autonomous Region, China (81°E, 43°N and 86°E, 44°N), but these ranges were not included in the paper.
2. The author only calculated the lowest daily temperature in the last 17 years, which could not guarantee there might be some exceptions surviving in those regions where the minimum daily temperature was below -34°C 17 years ago. This point should be discussed in the Discussion section.
3. In the Table 2, EAB was detected in Manitoba Winnipeg, Canada in 2017, but not detected in other years (2004, 2005, 2007, 2008, 2009, 2013, 2014, 2015, 2016, 2019), however, EAB still might be surviving in these years in the same location. This point also need to be discussed, especially on the detection methods, e.g., using color trap or pheromone trap were not good enough compared to girdling trap trees.
4. Althugh the absolute supercooling point of EAB is -35.3°C, but it overwinters inside the tree trunk where actual temperature is much higher than the air temperature outside. Thus, the conclusion on EAB can not survive in those regions below -34°C may not reliable.
Author Response
- Thank you very much for the valuable comments.
- In the "Materials and Methods", it was described that "……Asia (from 20 to 60 °N and from 100 to 150 °E) and……These squares covered the whole current range of the pest in the world and all of Europe". Emerald ash borer was found to occur in Xinjiang-Uygur Autonomous Region, China (81°E, 43°N and 86°E, 44°N), but these ranges were not included in the paper.
- Thank you for this good question. To clarify this we added the following explanation to the manuscript: Line 96: “We do not take into account the record from Xinjiang-Uygur Autonomous Region, China, since the number of found specimens and (or) damaged trees are unknown [4]. It is unclear if A. planipennis has established in this region”. Honestly, I do not believe in this record.
- The author only calculated the lowest daily temperature in the last 17 years, which could not guarantee there might be some exceptions surviving in those regions where the minimum daily temperature was below -34°C 17 years ago. This point should be discussed in the Discussion section.
- We have clarified this point in the Discussion section. Lines 234-237: We used the data since 2003, because A. planipennis was first recorded in Europe 2003. Little is known about the distribution of A. planipennis in Asia before its records in North America (2002) and European Russia (2003). The intensive studies of the native range began in 2003. There are no evidences that A. planipennis could survive the temperatures below -34°C in its native range before 2003.
- In the Table 2, EAB was detected in Manitoba Winnipeg, Canada in 2017, but not detected in other years (2004, 2005, 2007, 2008, 2009, 2013, 2014, 2015, 2016, 2019), however, EAB still might be surviving in these years in the same location. This point also need to be discussed, especially on the detection methods, e.g., using color trap or pheromone trap were not good enough compared to girdling trap trees.
- We agree that the case of Winnipeg is really very interesting. We have contacted Fiona Ross (Forestry Branch Manitoba Agriculture and Resource Development, Canada) about Winnipeg and have received the most current and the most reliable information from her (Lines 163-165) . She said, that planipennis was first detected in Winnipeg in November 2017. Then in January 2019 the temperature was –35.36 °C there. And then in summer 2019 no adults of A. planipennis were detected in Winnipeg. Therefore, it is unknown whether the population is stable or transient. It is quite possible, that A. planipennis was introduced to Winnipeg in 2016-2018, so it would be speculative to suggest, that the population survived the winters of 2004-2016. The comparison of different detection methods was not the goal of our study.
- Althugh the absolute supercooling point of EAB is -35.3°C, but it overwinters inside the tree trunk where actual temperature is much higher than the air temperature outside. Thus, the conclusion on EAB can not survive in those regions below -34°C may not reliable.
- Thank you for this comment. We corrected the text to clarify that the temperature inside the tree trunk cannot be much higher than the air temperature outside. (Lines 116-121). Venette and Abrahamson have studied the beneathbark temperatures of ash trees [https://www.nrs.fs.fed.us/disturbance/invasive_species/eab/control_management/cold_hardiness/bp-EAB-and-extreme-cold.pdf]. They write: “The bark does offer some insulation to the larva from nightly low temperatures. When these lows occur, temperatures beneath the bark are typically about 1°C (about 2°F) warmer than nearby air temperatures.”
- Thank you very much again. We wish you good health.
Round 2
Reviewer 1 Report
Line 222: I wouldn’t say these results apply to the upcoming 2 decades, because we don’t know what the world response to climate change will be, therefore the effects could be greater or less based on human activity. Also, rather than just writing what I wrote in the review, the authors really should read some literature about extreme events and add some citations.
Although I would never attempt to write a journal article in another language, and I very much respect anyone who attempts to do so, some editing will be necessary to improve the English. Particularly the use of “colds” when it should be “cold”. For example extreme “colds” and winter “colds”. But that is easily fixed.
Author Response
Thank you very much for the second review.
Line 222: I wouldn’t say these results apply to the upcoming 2 decades, because we don’t know what the world response to climate change will be, therefore the effects could be greater or less based on human activity. Also, rather than just writing what I wrote in the review, the authors really should read some literature about extreme events and add some citations.
- The literature on global climate change is extensive and contradictory. Some experts believe that this phenomenon is caused by human activity, others believe that only 4% of warming is because of humans, while other 96% is due to natural cyclic processes in the atmosphere. The change of the mean temperature of the coldest day in the previous 20 years was different in different regions of Europe, and we have no data on these changes in each region. We also have no prognosis of such changes for each region in the next 20 years. So our hypotheses about it would be speculative and amateurish. We just honestly state that our model does not take into account the global climate change (line 119).
Although I would never attempt to write a journal article in another language, and I very much respect anyone who attempts to do so, some editing will be necessary to improve the English. Particularly the use of “colds” when it should be “cold”. For example extreme “colds” and winter “colds”. But that is easily fixed.
- Thank you very much for your kind words. It is really not easy to write scientific articles in the foreign language. The English editing of our initially submitted manuscript was made by American Journal Experts. Then we made the scientific correction according the reviewers comments. We have no financial possibility to pay for the second English editing. So we did our best to correct the text ourselves according to your advices. In particular, “colds” is changed to “cold” and “temperatures” to “temperature”. The sentence “It is estimated that planipennis will eventually decimate nearly all ash (Fraxinus spp.) in North America.” is copy-pasted from DeSantis et al., 2013. Sorry, I don’t understand, what is wrong about “J-larva”.